# Optimization of Bit Allocation for Spatial Multiplexing in MIMO VLC System with Smartphones

**DOI:** 10.3390/s22218117

**Published:** 2022-10-23

**Authors:** Chang-Ming Lee, Bo-Hung Li, Chang-Chin Chiang

**Affiliations:** 1Department of Communications Engineering, National Chung Cheng University, Chiayi 621301, Taiwan; 2Advanced Institute of Manufacturing with High-Tech Innovations, Chiayi 621301, Taiwan

**Keywords:** optical channel, modulation, water filling algorithm, bit allocation

## Abstract

This paper focuses on the hardware components of smartphones, namely, the use of displays and cameras in mobile devices as transmitters and receivers to establish a near-field multiple-input–multiple-output (MIMO) visible light communication (VLC) system. Based on the relationship between the grayscale values of transmitted and received signals, the physical channel responses are detected and approximated with a high-order regression to obtain the channel gain. With the constraint of bit numbers in the MIMO VLC system, an integer-type water-filling scheme was designed for bit allocation to improve transmission efficiency. The physical examinations show that bit error rate (BER) reduction can be 26.4% with Gaussian noise of 30 dB and detected channel gain compared with the equal bit allocation. The optimization of the simulation was confirmed with the bit assignments in real cases.

## 1. Introduction

Recent decades have witnessed over a tenfold increase in smartphone traffic. The most critical issues with the ongoing development of mobile networks can be addressed by either increasing bandwidth or constructing more base stations to achieve the required throughput. These resources are rarer and more expensive during this phase of the rapid development and explosive growth of the wireless network. The International Telecommunication Union (ITU) predicted that the exponential growth of global mobile data traffic will continue and may reach about 60 zettabytes per year by the end of this decade [1]. Several data-intensive applications including autonomous vehicles, tactile Internet, virtual reality, augmented reality, and holographic projection could drive these expensive traffic demands, which can hardly be offered by the fifth generation of cellular technology (5G) [2]. Consequently, there is a growing research interest from both academia and industry towards B5G (beyond 5G), or the sixth-generation of mobile communications. The extensions of the spectrum [3], and the antenna array scale [4] can provide more spatial diversity and available spectrum resources. With this trend, numerous studies on artificial intelligence, orbital angular momentum multiplexing, visible light communication (VLC), etc., are dedicated to the development of 6G mobile communication systems [5].

Furthermore, VLC can increase the wireless transmission throughput, without increasing the licensed bandwidth or constructing high-cost base stations. Therefore, the ongoing requirements faced by the wireless industry are achievable. Multiple-input–multiple-output (MIMO) is famously used in the most fascinating wireless access technology to deliver the needs of the third-generation wireless mobile telecommunication technology (3G) and successors. MIMO technologies can integrate antennas, radios, and spectrums into a wireless access network to enable a higher capacity and speed for wireless communication. In advance, a massive MIMO can increase the throughput and spectral efficiency for emerging wireless standards because of the considerable array gain that a massive MIMO can achieve with a large number of antennas.

A basic VLC system is developed with the integration of a controller and optical components based on light-emitting diodes (LEDs) that deliver both lighting and data transmission. At the receiver part, a photodiode (PD) can detect the optical signals. Accordingly, the overall system design is inexpensive compared to radio frequency (RF) systems. In [6], some advantages of VLC include unregulated and license-free bandwidth, frequency reuse in adjacent rooms, security as light signals are undetectable behind walls, fewer health issues, and functionality in such environments where traditional RF systems are limited or forbidden.

In recent years, related developments vary not only in long-distance transmission via conductors or cables but also in near-field wireless communication. An increasing number of communication applications are especially used in mobile platforms, such as in smartphones and wearable devices. Due to advances in software and hardware designs, the optical components of mobile platforms have also been improved for high-value displays and cameras. Therefore, optical wireless transmission with mobile devices has emerged from radio frequency communication. This novel communication method can transmit information through visible light channels by converting electrical signals into intensity signals [7]. Therefore, optical components, such as a liquid-crystal display (LCD) and light-emitting diodes (LEDs), commonly designed for monitoring and illumination, are used as transmitters for VLC technologies. Moreover, the imaging capture components, including the complementary metal–oxide–semiconductor (CMOS) and charge-coupled device (CCD), are potential solutions for optical receivers. Generally, the advantages of VLCs [8,9,10] are safety, illumination, low cost, no electromagnetic interference, and a license-free spectrum. Since frequency band competition is increasingly fierce, VLC has become increasingly important.

To realize VLC techniques, precoding and equalization are critical for adapting the transmitter to the channels. Moreover, the joint design of the precoder and equalizer for a MIMO VLC system [11] usually suffers from a degraded performance in the low signal-to-noise ratio (SNR) regime. In preliminary research, the power allocation of a single-user MIMO VLC was studied with a design of modulation, multiplexing, and detection, as in [12]. Moreover, zero-forcing multi-user precoding techniques are also realized to improve VLC systems [13]. However, this incorporation of precoding and equalization developed in simulation cannot be directly applied to mobile devices because display-camera VLC systems utilize a low operating frequency (generally 30 Hz for display and capture). Moreover, the configurations of cameras, mainly adjusted by a mobile system (BIOS, firmware, or operating system), may yield significantly different signals and models. In the physical channel, transmission quality as bit error rate (BER) varies due to the issues of the mediums in channels, such as fogging or blurry conditions. Therefore, the optimization of communication systems, such as the water-filling (WF) method [14], can modify channel equalization with the power or quantity of bits to be transmitted. In the MIMO VLC system, each available channel has its maximum capacity. With the limitation of whole power according to the variety of channel capacity, the available power of channels can dominate the transmission efficiency [15,16] because the available power varies depending on various conditions, including interference and noise. Therefore, the management of signal powers for MIMO channels is one interesting aspect for improving the performance of the MIMO VLC system.

Based on the properties of VLC, most systems consider the intensity modulation and direct detection (IM/DD) to realize data communication. In [17], a direct-view MIMO VLC was constructed with a nonlinear optimization in the receiver. Similar to the WF method, the channel with maximal SNR has limitations in transmission power. However, the characteristics of mobile devices are quite different when compared with the conventional MIMO VLC systems. Issues surrounding signal power, response delay, system configuration, and data digitization are still open in the design and implementation. This paper proposes several techniques to construct a more practical system with an appropriate channel model and effective bit allocation for the modulation design.

## 2. System Model

The proposed MIMO VLC system is constructed with *N*_t_ emitters and *N*_r_ receivers, as shown in Figure 1. Generally, the modulation designed for VLC is compatible with spatial modulation (SM) and spatial multiplexing (SMP). Moreover, the modulation can be improved by optimizing power allocation for precoding and equalization. Compared with SM, SMP can increase the channel capacity or improve the transmission efficiency of communication systems. Although this scheme may suffer from the risk of inter-channel interference (ICI), the equalizer in the MIMO VLC system with spatial diversity [18] can overcome this issue and avoid a decrease in transmission efficiency. Moreover, the SMP can easily fit the bit allocation strategy designed with precoding and equalization in our system to maintain the transmission quality in each channel and improve the system throughput.

### 2.1. VLC Modulation

The transmitter is designed in a smartphone, as shown in Figure 2. Unlike the optical components used in traditional VLC systems, such as LEDs, PDs, etc., the smart device consists of an LED array display as the transmitter and a camera as the receiver. These optical components can be assembled as many transceivers due to their use of high-resolution devices. Consequently, the massive MIMO can be realized with these optical components, while the issue of ICI can be overcome.

SMP is one of the spatial diversity schemes for MIMO techniques with the same numbers of transmitters and receivers, such that both *N*_t_ and *N*_r_ are *N*. Each pair of transmitter and receiver can provide one channel for a data stream, as shown in Figure 3. In this case, four data streams are simultaneously launched. In [19], the spectrum benefits of SMP and SM with the same modulation order *M* can be obtained by (1) and (2), respectively. With a fixed number of channels and pulse-amplitude modulation (PAM) levels, SMP provides higher spectral efficiency compared to SM.
*N* × *log*_2_*(M)*(1)
*log*_2_*(N)* + *log*_2_*(M)*(2)

An optical channel with IM/DD is assumed. The signal **x** = [x_1_, x_2_, …, x*_N_*] is transmitted in the channel and the received data are formulated as:**y** = **Hx** + **W**(3)
where **H** is the channel matrix, and **W** is the sum of thermal noise and ambient shot light noise, which is independent of the transmitted signal, as indicated in [20]. Therefore, the entries of **W** are composed of a zero-mean additive white Gaussian noise (AWGN) in real numbers with the following variance:
(4)$$\sigma_{W}^{2}{=}_{\sigma thermal}^{2}+\sigma_{shot}^{2}$$
where $\sigma_{thermal}^{2}$ and $\sigma_{shot}^{2}$
are variances of thermal and shot noises, respectively. The channel matrix **H** can be expressed as:
(5)$$H=\left[\begin{array}{ccc} h_{1,1} & \cdots & h_{1N}\\ \vdots & \ddots & \vdots \\ h_{N,1} & \cdots & h_{N,N} \end{array}\right]\ldots$$
where *h_i_*_,*j*_ is the wireless channel transmission coefficient between the *j*-th transmitter and the *i*-th receiver. MIMO systems usually use the system model defined by (6) to indicate the channel gain. Light of sight (LOS) paths can be modeled with a generalized Lambertian radiant intensity [21]. In general, the energy of the reflected light can be omitted compared with the LOS light, and the system performance mainly depends on the optical signals in LOS [22]. The gain *h_i_*_,*j*_ in channel matrix **H** is:


(6)
$$h_{i,j}=\left\{\begin{array}{cc} \frac{\left(k+1\right)A}{2\pi d^{2}}{\text{cos}}^{k}\left(\varphi \right)\cos\left(\psi \right) & 0\leq \varphi \leq \psi_{\frac{1}{2}}\\ 0 & \psi >\psi_{\frac{1}{2}} \end{array}\right.$$


As shown in the Figure 4, *φ* is the angle of emergence to the emitter axis, and ψ is the angle of incidence to the receiver axis. The parameter *k* can be calculated with logarithmic terms as:
(7)$$\frac{-ln\left(2\right)}{ln\left(\cos\left(\Phi_{1/2}\right)\right)}$$

Φ_1/2_ is the half angle of the transmitter, Ψ_1/2_ is the half angle of receiver’s field of view (FOV), *A* is the detection area of the receiver, and the channel coefficient *h_i_*_,*j*_ depends on the position of the *j*-th transmitter and the *i*-th receiver. If the receiver and transmitter are not within the other’s coverage angle, then the gain is 0. Eventually, the channel matrix **H** considered in this paper can be measured in a physical environment. In the SMP system, the *M*-ary PAM with grayscale mapping is employed, and each intensity can be expressed in (8). The *M*-level PAM with grayscale intensity equation is:
(8)$$I_{m}=\frac{I}{M}m,m=1,\ldots M$$
where *I* is the maximum grayscale in the channel.

### 2.2. Detection of Physical Channels

The MIMO VLC communication system is constructed with two mobile devices for transmitters and receivers. For each pair of transmitter and receiver, the signals would be represented as grayscale values. The real channel in the MIMO VLC system needs to be detected with varying defects in the optical components and medium. In particular, the configuration of the camera may dominate the design of the receiver in the system. Therefore, the powers of signals used in this system are calculated with statistical tests in real cases. Firstly, the original signal in the transmitter is defined as an image block where each pixel has the same grayscale value according to the modulation level in each channel. Secondly, the area of interest block in the image captured by the digital camera (as receivers) is shrunk to half of the range located with the corner detection [23]. In the third step, all grayscale values in the same area are averaged as the received signal in the channel. In the practical MIMO VLC system, the analysis of each channel can provide the gain **H**, seen in (5), to perform the incoming bit allocation to improve the transmission efficiency.

### 2.3. Power Allocation

Because of the various channel conditions, the design of modulation used in the precoders and equalizers is essential for transmission performance. The power allocation needed to assign the modulation levels considering the capacities of all channels is critical for the transmission efficiency. While interference and noise are detected, the available quantity of transmission power can be precisely assigned. Moreover, the peak-to-average power ratio (PAPR) can be lowered to avoid power degradation. In Figure 5a, the channel with the maximum quantity of capacity is the first candidate to assign power until the boundary of channel capacity, while the greedy algorithm (GA) is applied for channel selection according to the remaining power. After the first power assignment in the best channel, the GA can then process the remaining power for the channel in the second position. This procedure continues until no channel is available or no power remains to be assigned. However, GA-based power allocation may cause an extreme imbalance of channel utilization and a lack of channel diversity. To allocate power in available channels as much as possible, power allocation based on the water-filling (WF) method [18] can provide more fairness and utilization of the MIMO system. To achieve a transmission diversity with fair utilization in MIMO channels, a WF-based scheme can granularly allocate power in more channels with an equal level of total energy (including powers of signal and noise), as shown in Figure 5b. In general, the maximum quantity of transmission efficiency is presented for the best gain of the channel with the lowest noise.

## 3. WF-Based Bit Allocation

To apply MIMO VLC techniques in mobile devices, the design of the power allocation techniques needs to consider the truth of the system and channel. Moreover, the configurations of optical components (displays and cameras) almost controlled by an operating system (BIOS or firmware) may change the channel conditions. These kinds of instability may cause gaps in power allocation between the simulation and the practical test.

Instead of power or energy values used in the simulation [17], bits are more feasible for the effective design of modulation to satisfy the requirements of digital communication in real MIMO VLC systems. Therefore, the objective and subjective formulations in the optimization of power allocation need to be modified in terms of bits. Based on the prototype of the WF algorithm, the proposed bit allocation for the MIMO VLC system can allocate more bits in better channels, and fewer bits for the worse channels. Following the results of the bit allocation, the design of the precoder in the transmitter and the equalizer in the receiver are simultaneously adjusted for all channels. Thus, *M*-PAM can be realized with the assigned bits in each channel. Moreover, optimization can be achieved by maximizing the channel diversity in the MIMO VLC system.

Similar to the optimization of transmission efficiency, the BER reduction is more feasible with the constraints of a fixed number of total bits transmitted in MIMO channels. Therefore, the goal of bit allocation for the design of precoding and equalization can be equivalent to the minimization of the error rate with a fixed total transmission rate. The objective and subjective functions for the scenario with *N* channels are respectively expressed as:
(9)$$\;\min\left\{\frac{1}{N}\sum_{k=1}^{N}\frac{I_{k}}{2\times \;2^{R_{k}}}\right\}$$
and
(10)$$\;\;\;\;\;\;\;\;\;\overset{N}{\underset{k=1}{\sum }}R_{k}=NR$$
where *I_k_* is the maximum grayscale in the *k*-th channel, *R* is the expected transmission rate per channel, and 
$\frac{x_{k}}{2\times 2^{R_{k}}}$
is the expected value of noise energy, as discussed in [24]. The Lagrangian equation *L**(R,λ)* can be formulated as:
(11)$$\;\frac{1}{N}\overset{N}{\underset{k=1}{\sum }}\left(\frac{I_{k}}{2*2^{R_{k}}}\right)+\frac{\lambda }{N}\left(\overset{N}{\underset{k=1}{\sum }}R_{k}-NR\right)$$
where ***R*** = (*R*_1_,*R*_2_, …, *R**_N_***) is the set of transmission rates for all channels. The error minimization can be solved by the following steps.

**Step 1:** Provide a Lagrange multiplier λ to enforce the expected number of transmission rates;

**Step 2:** Find the differentiation for each *R_k_* and λ;

**Step 3:** Solve the derived equation for each *R_k_* and then enforce the average transmission rate to fit the power constraint.

The first differentiations of the Lagrangian in terms of *R_k_* and *λ* with respect to zeros can be (12) and (13), respectively.


(12)
$$\frac{\partial }{\partial R_{k}}\left\{\frac{1}{N}\sum_{k=1}^{N}\left(\frac{I_{k}}{2*2^{R_{k}}}\right)+\frac{\lambda }{N}\left(\sum_{k=1}^{N}R_{k}-NR\right)\right\}=0,\forall k∊\left\{1,\ldots ,N\right\}$$



(13)
$$\frac{\partial }{\partial \lambda }\left\{\frac{1}{N}\overset{N}{\underset{k=1}{\sum }}\left(\frac{I_{k}}{2*2^{R_{k}}}\right)+\frac{\lambda }{N}\left(\overset{N}{\underset{k=1}{\sum }}R_{k}-NR\right)\right\}=0$$


These equations can provide the solution of the minimum transmission error by finding an appropriate set **R’** of power (rates) allocation {*R’_k_* | *k* = 1, …, *N*}. In advance, the maximal transmission rate *R’_k_* per shot in the *k*-th channel can be expressed as:
(14)$${R^{\prime }}_{k}=C+\frac{1}{2}log_{2}I_{k}$$
where


(15)
$$C=log_{2}\frac{ln2}{2\lambda }$$


Considering (14), the sum of all *R’_k_* must be *NR* such that the total signal power in the left term of (10) can be


(16)
$$\overset{N}{\underset{k=1}{\sum }}\left(C+log_{2}I_{k}\right)=NR$$


Thus, the channel capacity can be derived as:


(17)
$$C=R-\frac{1}{2}log_{2}{\left(\overset{N}{\underset{k=1}{\prod }}I_{k}\right)}^{\frac{1}{N}}$$


Considering (14), the bit allocation can be


(18)
$${R^{\prime }}_{k}=R-\frac{1}{2}log_{2}{\left(\overset{N}{\underset{k=1}{\prod }}I_{k}\right)}^{\frac{1}{N}}+\;log_{2}I_{k}$$


However, the rate *R_k_* provided by the WF algorithm is floating. Instead of direct rounding, the bit-level optimization can satisfy the requirement of digitization for communication or signal processing. In the proposed Algorithm 1, the output $R_{k}^{\mathrm{bit}}$
is the best bit allocation in the *k*-th channel. The allocation of the bit-wise process can evenly assign the desired bits to the most available channels with the aim of fewer transmission errors. Therefore, the channel diversity is realized in this MIMO VLC system.

**Algorithm 1.** WF-based bit allocation algorithm.1: Initialization: Bp=∑k=1NR′k; Rbit=[R1bit ,R2bit ,…,RNbit]=[0,0, …0]; 2: Calculate  R′k=R−log2(∏k=1NIk)1N+log2Ik for each *k*3: **While**
Bp>0 , **do** 4:  *k*’ = arg max(R′k|k=1,…,N) 5:  If R′k′< 0 then break and output failure;
5:  R′k′ = R′k′  − 1;  6:  Bp=Bp−1; 7:  Rk′bit=Rk′bit+1; 8: **end while**9: Output: Rbit


## 4. Experimental Results

This system is constructed with two Google Pixel 4a smartphones which are the transmitter and the receiver, respectively. The list of configurations is in Table 1. The frame size of 720 × 1520 is the same as the display size of the Google Pixel 4a. The system architecture is shown in Figure 6. Two mobile devices are mounted on a platform and are at a fixed distance (15 cm). The transmitter (display) is in the field-of-view of the receiver (camera). The centers of the display and camera are aligned. There are four transmitters and four receivers in the proposed 4 × 4 (*N* = 4) MIMO VLC system. Moreover, the signal transmitted by the VLC scheme is non-negative and could be transmitted in the wireless optical channel with IM/DD. The paradigm of image frames containing signals, captured by digital camera and segmented with a quarter area detected by the corner detection, are shown in Figure 7. These four blocks tagged with red rectangles are receivers. Instead of signal powers, the received data are the averaged grayscale of pixels in each block. This system can maintain spatial diversity due to a lack of joints of blocks and a lack of interference from other channels. In this example, the noise or interference in each channel is caused by the imperfect conditions of display, lenses, or air quality. If the system had neither precoding nor equalization, the communication quality would be severely worsened. Moreover, the signals in each channel may perform as grayscale values (integers in [0, 255] for 8-bit pixels) with *M*-PAM, and the maximum grayscale (*I*) is 255. Therefore, this study aims to improve transmission performance by adjusting the level of PAM to appropriately allocate bits for each channel based on the WF algorithm.

Before the bit allocation, the gain in each channel should be determined by considering the mapping of signals from transmitters and receivers. In Figure 8a, the transmitted signals are integers in the range [0, 255], and the received signals are floating-point in the same interval. With the regression, the high-order curve shown in Figure 8b can illustrate the channel gain.

With the equal bit allocation, the transmission bit number in each shot is two in each channel. Accordingly, each channel is incorporated with 4-PAM. Indeed, the proposed VLC system applies grayscale values to transmitted and received signals and is still in the same range [0, 255]. Therefore, the minimum transmitted signal is 0, and the maximum is 255. Regarding the gain of each channel, the received signals, in terms of averaged grayscale values, are shown in Table 2. The minimum mean-squared error (MMSE) equalizer is applied to evaluate the proposed system with AWGN, as noted in (3), similar to ambient shot light noise and thermal noise. In a real case with 30dB SNR (signal-to-noise ratio) and eight bits fully assigned for four channels, the BER of the MIMO VLC system with an equal bit allocation scheme [2, 2, 2, 2] is 0.1303, as shown in Table 3. The optimal allocation with the conventional WF method is [1.3, 2.6, 1.5, 2.6]. To complete the bit assignment, the rounded result [1, 3, 2, 3] cannot satisfy the total number of bits. With the proposed bit allocation strategy, the integer-type optimization *R^opt^* = [1, 3, 1, 3] can achieve the goal of eight bits in total. Therefore, the levels of PAM in each channel can be 2, 8, 2, and 8. The optimized average BER in the physical environment is 0.0959. Thus, the reduction in BER is about 26.4%. To verify the best case in both the simulation and real cases, the three best candidates obtained by a heuristic approach in the simulation are realized. The optimization is evident and identical in both simulation and real cases.

## 5. Conclusions

The proposed MIMO-VLC system consists of a display and camera in smart mobile devices as the transmitter and receiver. The basic arrangements comprise modulations (PAM and SPM), the detection of channel conditions, and power allocation. These designs are discussed with the characteristics of optical components in mobile phones. Moreover, the proposed WF-based bit allocation possesses the appropriate design for modulation in the MIMO VLC system. To evaluate this improvement, the optimization in the simulation is proven to have the minimum number of transmission errors in the physical cases. In future research, the proposed bit allocation for the MIMO VLC system can be extended for high-order PAMs and more dynamic channels for mobile platforms with higher resolutions for displays and cameras.

## Figures and Tables

**Figure 1 sensors-22-08117-f001:**
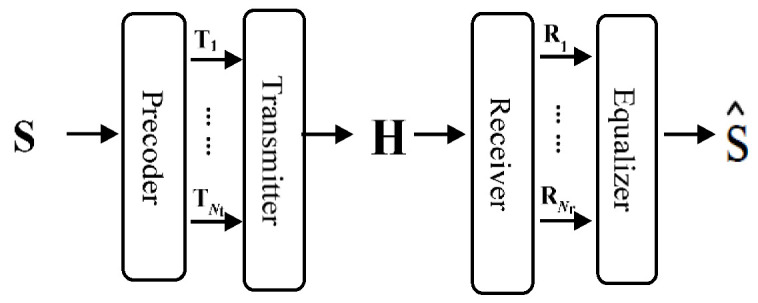
Diagram of MIMO VLC system.

**Figure 2 sensors-22-08117-f002:**
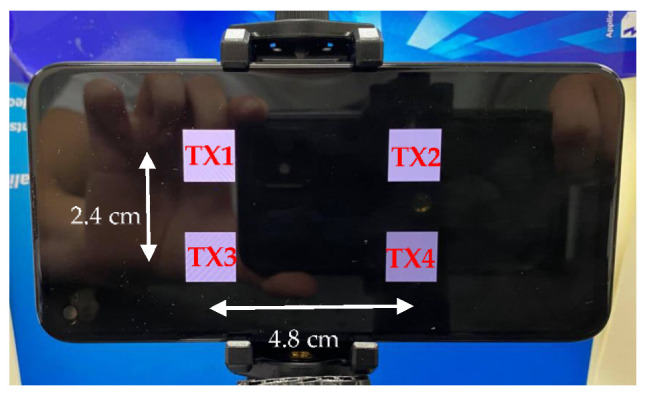
The paradigm of four transmitters in a 4 × 4 MIMO VLC system.

**Figure 3 sensors-22-08117-f003:**
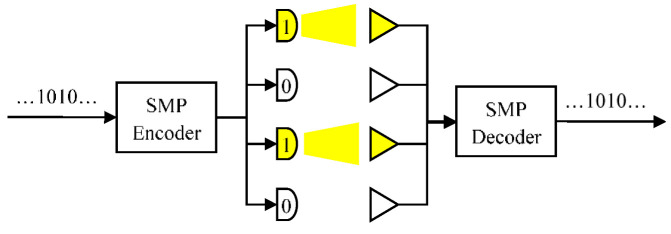
The paradigm of SMP with *N* = 4 and 2-PAM (pulse-amplitude modulation).

**Figure 4 sensors-22-08117-f004:**
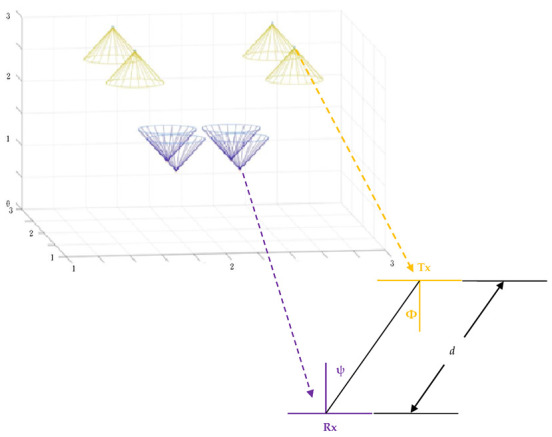
Geometric scenario (transmitters in yellow and receivers in purple) used to detect the channel response.

**Figure 5 sensors-22-08117-f005:**
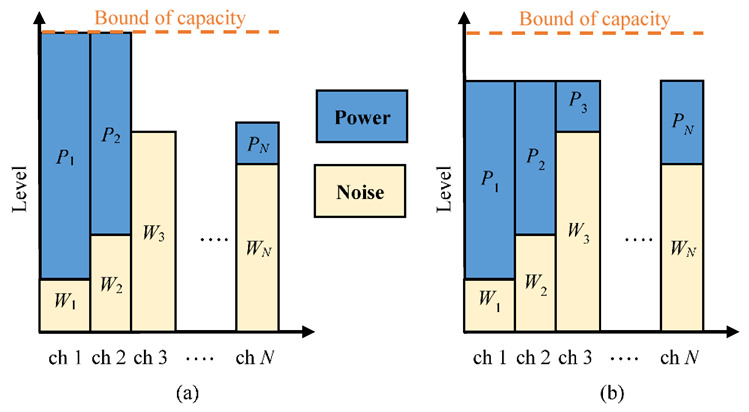
Paradigms of power allocation with (**a**) GA for one-by-one channel selection and (**b**) WF scheme for all channels with equal energy levels.

**Figure 6 sensors-22-08117-f006:**
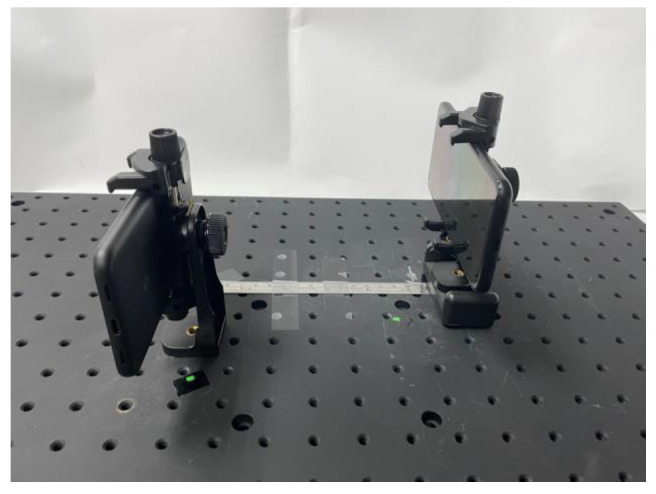
The VLC system architecture with two mobile devices mounted on the platform.

**Figure 7 sensors-22-08117-f007:**
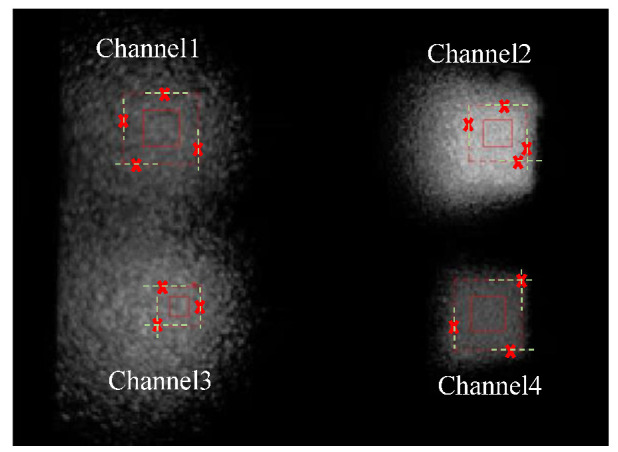
The paradigm of receivers (red rectangles) segmented with corner detection (noted with cross markers) and shrinking (from the dashed rectangles to solid ones) in the image captured by the camera.

**Figure 8 sensors-22-08117-f008:**
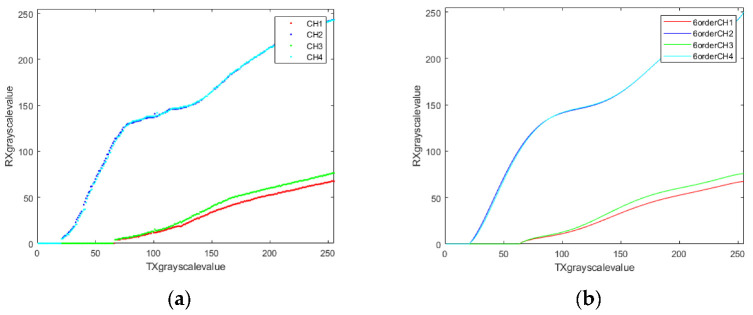
The relationships between transmitted and received signals are illustrated for four channels. (**a**) The pixel-wise detected models; (**b**) the high-order regression models.

**Table 1 sensors-22-08117-t001:** Configurations of the VLC system.

Parameters	Value
Transmission distance (cm)	15
Smartphone model	Google Pixel 4a
Frame rate on transmitter/receiver (fps)	30/29.6
Display pixel value (pixel)	720 × 1520
Number of transmitters	4
Number of receivers	4
Bits per frame	8
Camera resolution (pixel)	12.2 million
Smartphone chipset	Snapdragon 730 G
Smartphone CPU	2.2 GHz + 1.8 GHz, 64-bitmulti-core processor
Smartphone RAM	6 GB

**Table 2 sensors-22-08117-t002:** Coding of modulation levels with grayscale mapping in four practical channels.

Levels of PAM2/4/8	TX1	RX1	TX2	RX2	TX3	RX3	TX4	RX4
1/1/1	89	9.2	49	74.9	88	10.8	49	73.7
-/-/2	113	18.2	78	101.2	12	21	78	100.2
-/2/3	136	26.9	108	128.3	136	31.2	108	127.6
-/-/4	160	35.9	137	154.6	160	41.3	137	154.1
-/-/5	184	44.9	166	180.8	183	51.1	166	180.6
-/3/6	207	53.5	195	207.1	207	61.2	195	207.1
-/-/7	231	62.5	226	235.1	231	71.4	226	235.5
2/4/8	255	71.6	255	255	255	81.5	255	255

**Table 3 sensors-22-08117-t003:** Performance (BER) of simulation and physical case with different 4-channel allocations.

Bit Allocation	Simulation	Real Case
[2, 2, 2, 2]	0.0153	0.1303
[0, 3, 2, 3]	0.0048	0.1042
[2, 3, 0, 3]	0.0043	0.1182
[1, 3, 1, 3]	0.0010	0.0959

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
