# Peer review of "Optimization of Bit Allocation for Spatial Multiplexing in MIMO VLC System with Smartphones"

_sensors, 2022, doi:10.3390/s22218117_

Round 1

Reviewer 1 Report

I am sorry to tell that the article cannot be accepted in the present form. While the scientific content might be of interest, the English style is at a level at which the majority of the manuscript is incomprehensible. I could make a point by point list but basically every sentence has problems. I strongly suggest that the authors get some professional help to improve the overall writing before resubmission.

Author Response

Dear associate editor and reviewers:

Thank you very much for your reviewing our manuscript and giving useful comments. We have checked our manuscript and revised it carefully taking into account your valuable advice. Your comments and suggestions contributed to an improved manuscript. Here is our response to your specific comments:

Manuscript ID: Sensors-1920132

Title: Optimization of Bit Allocation for Spatial Multiplexing for MIMO VLC System in Smartphones

Authors: Chang-Ming Lee*, Bo-Hung Li, Chang-Chin Chiang

Moderate English changes:

This manuscript has been edited by MDPI English Editing (English-51613) and evaluated by the basic writing suggestions of “Grammarly” (https://app.grammarly.com/).

The responses to the rest issues in the list:

R1: Three paragraphs added to show the value of this study are about the background and relevant references provided in the first section (Introduction).

R2: The quality of the cited references is improved with modified illustrations and descriptions in a new order. In addition, there are more references ([1][3][4][5][11]) relevant to this research.

R3(about novelty): The concerns in this research are the properties of the platform (mobile phones), practical channel modeling and effective bit allocation for the modulation. The novelty of this study is to solve the issues about the implementation of the MIMO VLC system designed in real channels. The goals and difficulties are added in the last two paragraphs of the first section (Introduction) to provide more appropriate statements about this research.

R4: The misunderstanding about the proposed scheme is mainly caused by the weakness of descriptions. Most parts of the designs (including the proposed WF-based bit allocation algorithm) in Sections II and III are edited. Besides, several crucial statements (marked in yellow) are added in this revision.

R5: To present the results, several phases in Section IV are edited with more details. In particular, the descriptions of the receivers (about Fig.7) are shown in the first paragraphs. These relevant statements marked in yellow can be found in Section IV.

R6: In this revision, the modification of conclusions in Section V is completed with rewriting and editing. The results presented in Section IV can support our contributions.

Reviewer 2 Report

The authors propose a system that includes a screen and camera as the transmitter and receiver in smart mobile devices. In the current work by VLC, PAM and SMP constructed a MIMO VLC system.  However, the novelty of the present work is weak and not compared to other works, and an uncompleted introduction. In addition, there are many lacks in writing and presentation of the work, quality of figures and…..

Author Response

(The authors gave the same response as above.)

Round 2

Reviewer 1 Report

I am pleased to state that the manuscript has significantly improved since last review round, and it's now worth of being accepted for publication

Reviewer 2 Report

The authors relatively improved my demands and improved the presentation of their work to an acceptable extent, only the quality of some figures need to be better.